# Subtractive transformation of cathode materials in spent Li-ion batteries to a low-cobalt 5 V-class cathode material

Jun Ma[1], Junxiong Wang ®[1,2] ✉, Kai Jia[1,2], Zheng Liang ®[2], Guanjun Ji[1,2], Haocheng Ji[1], Yanfei Zhu[1], Wen Chen[1], Hui-Ming Cheng[3,4] ✉ & Guangmin Zhou ®[1] ✉

Adding extra raw materials for direct recycling or upcycling is prospective for battery recycling, but overlooks subtracting specific components beforehand can facilitate the recycling to a self-sufficient mode of sustainable production. Here, a subtractive transformation strategy of degraded $LiNi_{0.5}Co_{0.2}Mn_{0.3}O_2$ and $LiMn_2O_4$ to a 5 V-class disordered spinel $LiNi_{0.5}Mn_{1.5}O_4$-like cathode material is proposed. Equal amounts of Co and Ni from degraded materials are selectively extracted, and the remaining transition metals are directly converted into $Ni_{0.4}Co_{0.1}Mn_{1.5}(CO_3)_2$ precursor for preparing cathode material with in-situ Co doping. The cathode material with improved conductivity and bond strength delivers high-rate (10 C and 20 C) and high-temperature (60 °C) cycling stability. This strategy with no extra precursor input can be generalized to practical degraded black mass and reduces the dependence of current cathode production on rare elements, showing the potential of upcycling from the spent to a next-generation 5 V-class cathode material for the sustainable Li-ion battery industry.

Sustainable production of rechargeable Li-ion batteries (LIBs) is important to meet the ever-growing market demand for electric vehicles and portable electronics[1,2]. However, limitations will arise in the context of increasing production requirements, one is that the raw materials required for today's LIBs chemistry are scarce and rarely sourced sustainably, and the other is poor sustainability in battery manufacturing and spent battery recycling[3–5]. The cathode material is the key component with the highest mass and cost ratio in LIB production, and is the most valuable, especially for the one containing cobalt (Co)[6,7]. Transition metal layer oxide materials, e.g. $LiCoO_2$ (LCO), $LiNi_xCo_yMn_{1-x-y}O_2$ (NCMs), and $LiNi_{1-x-y}Co_xAl_yO_2$ (NCAs), one of the most consumed cathode materials in LIBs, are all Co-containing cathode materials. The ever-growing LIBs demand will inevitably

increase Co consumption. However, the low reserves and uneven distribution in the earth, unstable supply caused by mining, and high cost of Co present challenges affecting cathode production[8–10].

The development of next-generation cathodes with low-Co content is a possible solution to alleviate concerns about sustainable battery production due to the historically inelastic supply of Co[8]. Among which, the spinel $LiNi_{0.5}Mn_{1.5}O_4$ (LNMO) is an appealing cathode material for its robust structure with a three-dimensional ion channel[11], high operating voltage (~4.7 V versus $Li^+/Li$), and its use in high energy density LIBs[12,13]. The Mn-rich chemical composition of LNMO makes it more competitive compared with Ni-rich one considering the low cost and high reserves of Mn in the earth and low environmental impact[9]. However, the problem of transition metal

[1]Tsinghua-Berkeley Shenzhen Institute & Tsinghua Shenzhen International Graduate School, Tsinghua University, Shenzhen 518055, China. [2]Frontiers Science Center for Transformative Molecules, School of Chemistry and Chemical Engineering, Shanghai Jiao Tong University, Shanghai 200240, China. [3]Faculty of Materials Science and Energy Engineering/Institute of Technology for Carbon Neutrality, Shenzhen Institute of Advanced Technology, Chinese Academy of Science, Shenzhen 518055, China. [4]Shenyang National Laboratory for Materials Science, Institute of Metal Research, Chinese Academy of Science, Shenyang 110016, China. ✉e-mail: wjx1992@sjtu.edu.cn; hm.cheng@siat.ac.cn; guangminzhou@sz.tsinghua.edu.cn

dissolution and inferior high-temperature performance of LNMO hindered its application[14–16]. Finding an effective modification way to synthesize LNMO with improved performance is of great significance to the sustainability of the LIBs industry[13].

The sustainable reuse of degraded cathodes in spent LIBs is an effective means to relieve raw materials supply pressure and improve the sustainability of battery production[3,17]. As the amount of spent LIBs surges, recycling cathodes, especially for Co-containing ones is urgent. Among which, the $LiNi_{0.5}Co_{0.2}Mn_{0.3}O_2$ (NCM523) accounts for a considerable proportion of the market share in the last few years for its good chemical composition with a balanced ratio of Co to Ni and Mn, as well as its excellent comprehensive performance, e.g. comparable capacity output and better thermal stability to that of Ni-rich materials[18]. An emerging direct recycling method based on composition supplementation and structure repairing of cathode materials (e.g. LCO[19], NCM523[20], $LiFePO_4$ (LFP)[21,22], etc[23,24].) has been verified as an effective way for degraded NCM523 (D-NCM523) recycling. It defaults to an addition strategy requiring extra raw materials, additional lithium salt for the regeneration, or both lithium and nickel salts for upcycling from the low-Ni cathode material to the high-Ni one is commonly used[25,26]. The physicochemical composition and structure of recycled materials were restricted on the characteristics of intrinsic degraded cathode materials, showing limitations in lowering the Co amount and targeting different spent cathodes with various degradation levels[27]. Moreover, scale-up tests based on this route have not been well validated, it is still a promising but challenging recycling route[28].

In this work, we propose a simple and scalable recycling route for recycling D-NCM523/D-NCM523 black mass and degraded $LiMn_2O_4$ (D-LMO) to produce a low-Co doped LNMO (Co-LNMO), towards to solve the two key challenges affecting the sustainable production of batteries. It changes the default idea of addition strategy in the direct recycling route for transforming the spent to the repaired or the upcycled, instead, this is a subtractive strategy inspired by the idea: less is more (Fig. 1). Extra raw materials, such as lithium salt (e.g. $Li_2CO_3$ and LiOH) and nickel salt for the cathode regeneration or upcycling are no longer required. A selective precipitation process to preferentially subtract partial Co and Ni was designed. Given the important role of Co in the structural stability of cathode materials and the performance output of batteries, the modest amount of Co used in the future-oriented cathode is necessary and more reasonable[29]. Deliberately retaining Co in LNMO spinel structure is demonstrated to improve the conductivity and structural stability, delivering a high-rate and high-temperature performance of Co-LNMO. The problem of transition metal dissolution in the long-term cycling of LNMO especially under high-temperature working conditions is solved, which has been verified from the post-mortem analysis. In addition, this subtractive strategy not only meets the trend for developing next-generation low-Co cathodes but also facilitates the recycling process to a self-sufficient mode of sustainable cathode production, the lithium and transition metal elements from D-NCM523 and D-LMO are all reused, in line with the current development of closed-loop recycling.

## Results

### Subtractive transformation process and intermediate products

The molar ratio of Ni and Li in NCM523 is consistent with that in LNMO, making the transformation from D-NCM523 to LNMO possible without the addition of extra Ni and Li precursors. The D-NCM523 and D-LMO powders were harvested from degraded cathodes in spent batteries (see "Methods"), and D-NCM523 black mass from the battery recycling

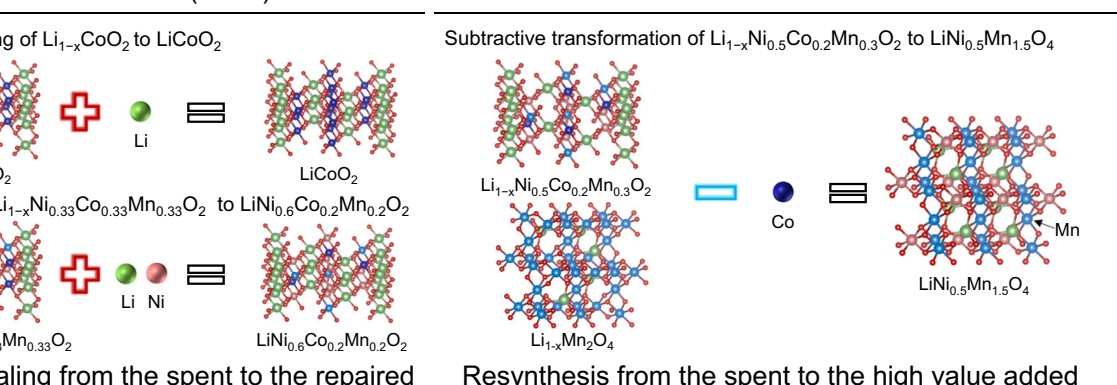
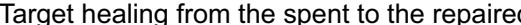
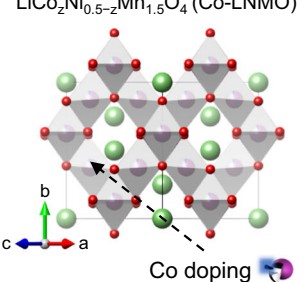

**Fig. 1 | Schematic of recycling strategy.** Schematic of the additive strategy for the direct recycling (A− to A)/upcycling (A− to A+) of spent cathode materials and the subtractive strategy for transforming different spent cathode materials to high-performance cathode materials (A & B to X). The working principle of the additive strategy relies on the lithium supplementation reaction during a high-temperature process, delivering a single type of regenerated cathode material. In contrast, the subtractive strategy makes the upcycling of complex solid waste towards a next-generation low-Co 5 V-class cathode material, an alternative idea for the sustainable use of resources.

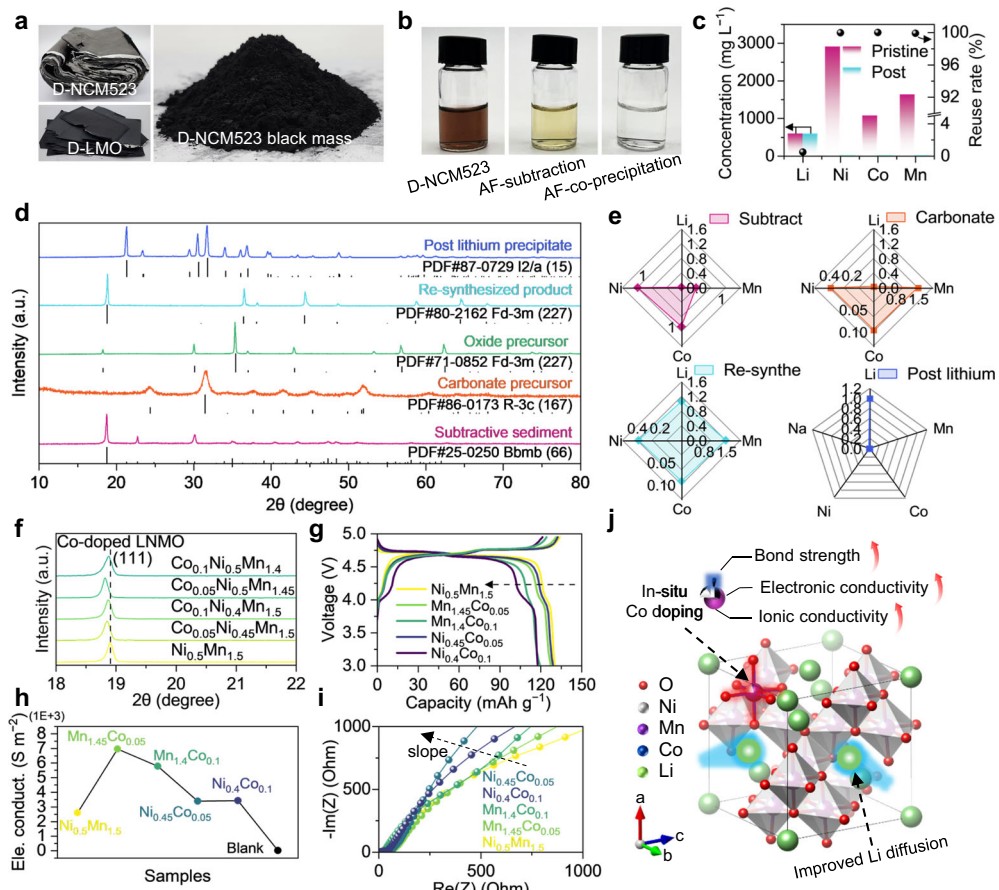

**Fig. 2 | Characterizations of intermediate products and Co-LNMO in the proposed subtractive transformation recycling process. a** Photos of spent cathodes and their material powders: D-NCM523, D-LMO, and the black mass of D-NCM523. **b** Photos displaying the composition change of the solution during the recycling process, "After" is shortened to "AF". **c** Element concentration of the pristine D-NCM523 solution and its solution after the co-precipitation process, and the corresponding reuse rate of every metal element. **d, e** XRD patterns and the relative Li/Ni/Co/Mn molar ratios of intermediate products including subtractive sediment, carbonate, and oxide precursors, re-synthesized product, and post lithium precipitate. A series of LNMO samples with Co substitution (0.05/0.1-mole ratios) of Ni or Mn: XRD patterns (**f**), charge and discharge curves (**g**), electrical conductivity comparison (**h**), and Nyquist plots (**i**) which could show the changing trend of the Li⁺ diffusion capability. **j** Schematic of the role of in-situ Co doping in spinel LNMO structure.

plant was also used (Fig. 2a). Transition metal elements with a highly oxidized state in the cathode materials were all reduced to "+2" ions after the dissolution process (Supplementary Fig. 1). The separation of $Ni^{2+}$ and $Co^{2+}$ ions is difficult from the E-pH diagrams[30]. To obtain an ideal stoichiometric ratio of each element for the direct preparation of the precursor, $(NH_4)_2C_2O_4$ solution was used for preferential subtraction of both $Ni^{2+}$ and $Co^{2+}$ ions from the dissolved D-NCM523 solution to form oxalate precipitate by making full use of the similar chemical behavior in the aqueous solution of both elements when the pH is low in acidic condition[31,32]. The residual ions remain in the solution due to complexation with $NH_4^+$ (Fig. 1). A carbonate $Ni_xCo_yMn_{1.5}(CO_3)_2$ (x + y = 0.5) precursor was obtained by a simple co-precipitation method with the addition of $Na_2CO_3$ after using the dissolved D-LMO solution for desired Mn stoichiometric ratio regulation. The $Li^+$ was recycled for the resynthesis of Co-LNMO (see "Methods"), which is a meliorative recycling strategy compared with the multi-step separation process of every single element in regular hydrometallurgy strategy (Supplementary Fig. 1 and Supplementary Table 1). The state of the solution changes significantly in the subtractive transformation recycling process (Fig. 2b). The transparent dark brown solution rich in metal ions was a pristine dissolved D-NCM523 solution, and the color of the solution changes to faint yellow because of the Co and Ni extraction. A colorless solution was obtained after the co-precipitation process, indicating the efficient reuse (>99.9% reuse rate) of the transition metal

ions (Fig. 2c). Besides, less than 0.5% of $Li^+$ was lost in the recycling process, indicating negligible lithium impurities in intermediate products.

Figure 2d shows the X-ray diffraction (XRD) patterns of the intermediate products. The subtractive sediment obtained in the subtraction process has an orthorhombic crystal structure with the *Bbmb* space group, corresponding to the XRD pattern $C_2CoO_4 \cdot H_2O$. The relative Li/Ni/Co/Mn molar ratios acquired by inductively coupled plasma optical emission spectroscopy (ICP-OES) of the subtractive sediment (Fig. 2e) show a nearly equal mole ratio of Ni and Co co-precipitation in the sediment due to similar precipitation characteristics of $Ni^{2+}$ and $Co^{2+}$ in the aqueous solution with $C_2O_4^{2-}$, few $Mn^{2+}$ and no $Li^+$ was precipitated. The carbonate precursor, oxide precursor obtained by the thermal decomposition of carbonate precursor, post lithium precipitate, and the re-synthesized product show well-matched crystal structure with the phase of references (Fig. 2d). The relative Ni/Co/Mn molar ratios of carbonate precursor and the re-synthesized product were 0.4, 0.1, 1.5 (Fig. 2e), indicating successful substitution of Ni by Co in the obtained product. The crystal structure of post-lithium precipitate matches well with monoclinic $Li_2CO_3$ without any impurities.

To investigate the Co substitution effects in spinel LNMO, different doping ratios of Co for replacing partial Ni or Mn were studied. The strongest characteristic peaks of all Co-doped LNMO samples showed

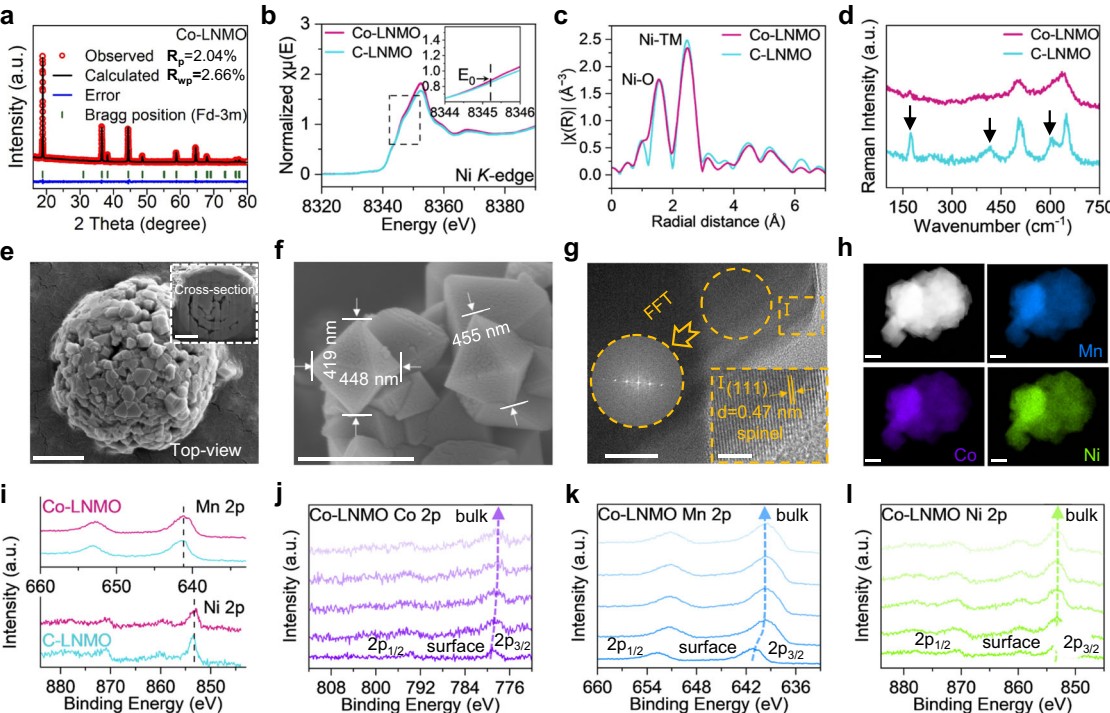

**Fig. 3 | Crystal structure, morphology, and composition of Co-LNMO. a** XRD Rietveld refinement of Co-LNMO. Ni K-edge XANES spectrum (**b**) and Fourier transform EXAFS spectrum (**c**) of Co-LNMO and C-LNMO. **d** Raman spectra of Co-LNMO and C-LNMO. **e** FIB-SEM image of Co-LNMO with a secondary particle structure (scale bar = 1 μm). Inset: Cross-section image (scale bar = 1 μm). **f** SEM image of a primary particle of Co-LNMO (scale bar = 500 nm). **g** TEM image of Co-LNMO (scale bar = 20 nm). Inset: HRTEM image of Co-LNMO from the selected area

(scale bar = 5 nm), FFT pattern from the circled area. **h** TEM-EDS maps of Co-LNMO and Mn, Co, Ni elements (scale bar = 200 nm). **i** XPS spectra of Mn 2p and Ni 2p of Co-LNMO and C-LNMO. XPS spectra of different elements at different depths in Co-LNMO: the Co 2p (**j**), the Mn 2p (**k**), and the Ni 2p (**l**). The signals were collected from the surface to the bulk after sputtering with an Ar⁺ cluster ion source, each signal acquisition interval is 5 min.

a very weak shift to the lower angles compared with LNMO (Fig. 2g). The detailed crystal structure information, e.g. atomic occupation and bond length, was revealed by XRD Rietveld refinement analysis (Supplementary Figs. 2a–e). The transition metal and oxygen (TM-O) bond lengths in the series of Co-doped LNMO samples were slightly shorter than that in pure LNMO, indicating an improved bond strength of Co-doped LNMO samples[33]. In particular, the sample with the 0.1-mole ratios Co substitution of Ni ($Co_{0.1}Ni_{0.4}$) showed lower bond strength than the samples with the 0.05-mole ratios (Supplementary Fig. 2f), indicating stronger TM-O bonds with the increase of Co substitution Ni in the spinel structure. The redox activities during the electrochemical process were influenced due to the Co substitution effect[34]. For the samples with 0.05 doping ratios, no matter whether Mn or Ni was substituted, there was no significant change in the charging and discharging plateau at ~4.7 V for $Ni^{2+}/Ni^{4+}$ or ~4.0 V for $Mn^{3+}/Mn^{4+}$ (Fig. 2g). For the samples with 0.1 doping ratios, more Co substitution leads to the obvious charge compensation effect, which reduced the amount of redox-active ions beneficial to capacity output contribution. Both $Co_{0.1}Mn_{1.4}$ and $Co_{0.1}Ni_{0.4}$ samples had shorter charging and discharging plateau at ~4.7 V, despite the longer plateau at ~4.0 V of $Co_{0.1}Ni_{0.4}$. The longer plateau indicated more $Mn^{3+}/Mn^{4+}$ redox couples, which may lead to an increase in conductivity for more trivalent ions and the mixed valence effect[35]. Although the capacity of all Co-doped LNMO samples showed ~5 mAh·g⁻¹ reduction to LNMO (Fig. 2g), they had improved structural stability and redox reversibility with improved cyclic stability, especially for the $Co_{0.1}Ni_{0.4}$ sample (Supplementary Fig. 3). Besides. the electronic conductivity of Co-doped LNMO cathodes was higher than that of pure LNMO (Fig. 2h), as well as the Li⁺ diffusion capability (Fig. 2i) which is proportional to the slope of the diagonal[36], making all Co-doped LNMO samples with a pronounced increase in rate performance (Supplementary Fig. 4). As the schematic

illustrated in Fig. 2j, Co substitution can improve the conductivity of LNMO and the strength of TM-O bonds in spinel structure[33], making Co-doped LNMO samples better rate performance and cyclic stability, especially for $Co_{0.1}Ni_{0.4}$ sample. Therefore, it is promising for the preparation of $LiCo_{0.1}Ni_{0.4}Mn_{1.5}O_4$ by the subtractive transformation recycling process.

## Characterization of Co-LNMO

The re-synthesized Co-LNMO was identified as single-phase. The XRD pattern of Co-LNMO was fully indexed using a cubic unit cell with $a = 8.1637(1)$ Å ($a = b = c$), and $V = 544.07(9)$ Å³ (Fig. 3a). Detailed structure information revealed by Rietveld refinement analysis showed Co-LNMO has slightly shorter $a$ and smaller $V$ compared with commercial LNMO (C-LNMO) (Supplementary Table 2). According to the Ni K-edge X-ray absorption near edge structure (XANES) spectrum, the Co substitution of Ni did not make the pre-edge change of Co-LNMO (Fig. 3b), suggesting the constant octahedral geometry of Ni after Co doping. In addition, the $E_0$ in the first derivative of the energy edge of Co-LNMO is identical to that of C-LNMO, indicating an unchanged oxidation state of Ni in the bulk environment (Supplementary Fig. 5). The $k^3$-weighted Fourier transform of extended X-ray absorption fine structure (EXAFS) spectrum was shown in Fig. 3c. The two coordinated shells are assigned to the Ni-O and Ni-TM scattering paths, respectively. The slightly weaker intensity of Ni-TM oscillation in Co-LNMO reveals its less ordered local structure. Raman spectroscopy, sensitive to the atomic-scale structure variation, was also used to reveal the short-range local environment of Co-LNMO[37]. Under the same test conditions, Raman modes of Co-LNMO were broadened and exhibited significantly lower intensities (Fig. 3d), indicating the cations in a more disordered manner. Less ordered phase in Co-LNMO can be verified from the lower intensities of $T_{2g}$ and $E_g$ modes at 160 and 410 cm⁻¹. The

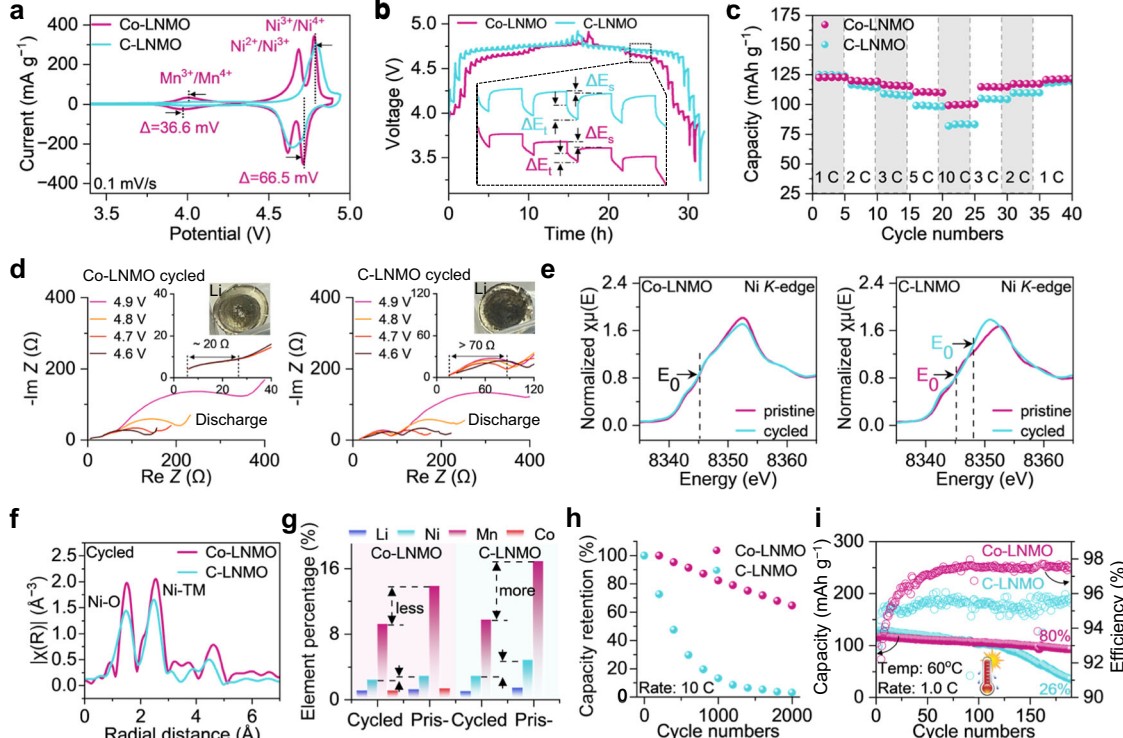

**Fig. 4 | Electrochemical characterizations and local coordinate environment of Co-LNMO. a** CV of Co-LNMO and C-LNMO in a coin cell. **b** GITT curves of Co-LNMO and C-LNMO at the same cycle number. Inset: enlarged plot in the discharge process in the GITT mode. **c** Rate capability measurements of Co-LNMO and C-LNMO in a constant-current (CC) mode, 1C = 146 mAh·g⁻¹. **d** In-situ EIS spectra of cycled Co-LNMO and C-LNMO after 400th cycling. Inset: enlarged plot of impedance change in the high-frequency area; photo of Li anode of the cycled cells. **e** Ni K-edge XANES spectrum of p5ristine and 400th cycled Co-LNMO and C-LNMO. **f** Ni K-edge Fourier transform EXAFS spectrum of 400th cycled Co-LNMO and C-LNMO. **g** Element percentage of pristine and 400th cycled Co-LNMO and C-LNMO, "Pristine" is shortened to "Pris-". **h** Capacity retention of Co-LNMO and C-LNMO during cycling at a 10 C rate at 30 °C for 2000 cycles. **i** Cycling performance of Co-LNMO and C-LNMO at 60 °C.

indistinguishability/widening of $A_{1g}$ modes of Mn⁴⁺ and Ni²⁺ at 580–620 cm⁻¹ indicates the lower concentration of Mn⁴⁺ and Ni²⁺ and the decrease of partial cation order on a local scale[38,39], which may deliver Co-LNMO with a higher electrical conductivity than the ordered one[37].

According to focused ion beam scanning electron microscopy (FIB-SEM), the Co-LNMO powder was a secondary spherical particle structure (Fig. 3e), consisting of predominant octahedral primary particles of 400–500 nm (Fig. 3f). A similar structure with smaller primary particles was seen in C-LNMO powder (Supplementary Fig. 6a–c). The transmission electron microscopy (TEM) images showed Co-LNMO consistent lattice fringes and smooth edges without lattice distortion (Fig. 3g), the lattice space of 0.47 nm corresponds to the (111) plane of the typical spinel structure, which also verified by the inset fast Fourier transform (FFT) pattern. The energy dispersive spectroscopy (EDS) maps showed Co-LNMO with uniform Co doping and distribution of transition metal (Mn, Co, Ni) elements. The effect of in-situ Co doping on other transition metals of spinel structure was investigated by X-ray photoelectron spectroscopy (XPS) and depth profiling analysis. For the XPS spectra of Mn and Ni 2*p*, there was no characteristic peak shift (Fig. 3i), indicating the unchanged oxidation state of both. Only the relative intensity of Ni 2*p* was lowered in Co-LNMO because of the partial Co substitution of Ni. The XPS spectra of Co at different depths in Co-LNMO confirmed the uniformity of Co substitution at both the surface and the bulk (Fig. 3j), and no satellite peaks were observed, indicating the oxidation state of Co (III) in the disordered spinel structure. The same trends for the shift of characteristic peaks to the lower binding energy were observed in the XPS

spectra of Co, Mn, Ni 2*p* (Fig. 3j–l), which were attributed to the reduction of oxidation state by Ar⁺ sputtering.

## Electrochemical characterizations and structure stability of Co-LNMO

The cyclic voltammetry (CV) curves (Fig. 4a) for the Co-LNMO cathode witness a more obvious plateau-like region at ~4.0 V with a ~36.6 mV hysteresis, indicating the slight increase of Mn³⁺/Mn⁴⁺ redox reaction due to Co-doping introduced charge compensation effect. Two pairs of sharp symmetric peaks at ~4.68/4.62 and 4.78/4.72 V with a lower peak-to-peak separation were observed in Co-LNMO, which can be attributed to the Ni²⁺/Ni³⁺ and Ni³⁺/Ni⁴⁺ redox reaction. In contrast, only a pair of peaks at ~4.80/4.65 V attributed to Ni²⁺/Ni⁴⁺ was observed in C-LNMO, indicating the lower cation ordering of Co-LNMO than C-LNMO[40], in good agreement with the previous Raman results. Figure 4b shows the voltage profiles of Co-LNMO and C-LNMO by the galvanostatic intermittent titration technique (GITT) test. The Co-LNMO has a smaller total voltage change ($\Delta E_t$) and bigger voltage evolution of the steady state for the related step ($\Delta E_s$) compared to C-LNMO, indicating the improvement of Li⁺ diffusion abilities[41]. Co-LNMO delivered a higher capacity than C-LNMO at a high rate (Fig. 4c) and showed better cycling stability than C-LNMO at 1C (Supplementary Fig. 7a). It retains >80% capacity retention after 400 cycles, better than that of C-LNMO (<75%). The improved cycling performance was also observed in Co-LNMO re-synthesized by D-NCM523 black mass (Supplementary Fig. 7b). In-situ impedance changes at discharging state of the cycled Co-LNMO and C-LNMO were also conducted (Fig. 4d). The solid electrolyte interface (SEI) resistances represented by the first

semicircle in the Nyquist plot in Co-LNMO cell kept stable (~20 Ω), even at different state of charge (SOC). The resistance of C-LNMO cell increased and fluctuated (>70 Ω) at different SOC, indicating an unstable SEI in C-LNMO cell after long-term cycling. The surface of the Li anode in the cycled Co-LNMO cell showed fewer byproducts than that in cycled C-LNMO (Fig. 4d), which further confirmed the formation of a stable interface in the cell by using Co-LNMO as the cathode.

The Ni redox at high voltage is the main reaction during LNMO cycling, which may be affected by the local environment variation of spinel structure and irreversible reduction of Ni ions after long-term cycling. The Ni K-edge XANES spectrum of cycled Co-LNMO showed less fine-structure change compared with the pristine Co-LNMO (Fig. 4e), the nearly same $E_0$ in the first derivative of the energy edge of both samples indicates an unchanged oxidation state of Ni (Supplementary Fig. 8). An obvious uplift of cycled C-LNMO at 8340–8344 eV relative to pristine C-LNMO illustrates the lattice distortion (Supplementary Fig. 9a), in contrast with a weak uplift in the pristine and cycled Co-LNMO showing better lattice stability (Supplementary Fig. 9b). The $E_0$ in the first derivative of the energy edge of cycled C-LNMO is larger than that of pristine C-LNMO (Supplementary Fig. 8), indicating the increased oxidation state of Ni in the bulk structure of cycled C-LNMO[42]. There were weaker intensities of Ni-O and Ni-TM oscillation in cycled C-LNMO than that in cycled Co-LNMO (Fig. 4f), indicating less ordered local structure as well as less coordination around Ni-O and Ni-TM sites in cycled C-LNMO, which is further confirmed by the 2D contour Fourier-transformed Ni K-edge EXAFS (Supplementary Fig. 10). The weaker Ni-O and Ni-TM oscillation and the lower corresponding **k** of cycled C-LNMO was observed, in sharp contrast to that in pristine Co-LNMO and C-LNMO (Supplementary Fig. 11), indicating the unstable Ni coordination environment after C-LNMO cycling. In contrast, there was a minor change in the intensity maximum and corresponding **k** between the pristine and cycled Co-LNMO (Supplementary Figs. 10 and 11), indicating the stable Ni coordination environment and good structure stability of Co-LNMO. No characteristic peak shift in XRD patterns of cycled Co-LNMO further confirmed a stable lattice in Co-LNMO (Supplementary Fig. 12). Due to that, transition metal dissolution in Co-LNMO was also alleviated (Fig. 4g). For the high-rate cycling performance, the Co-LNMO cycling at 10 C or 20 C rate for over 1000 cycles showed less than 20% capacity fading (Fig. 4h and Supplementary Fig. 13), indicating a complete structure transition with less stress concentration during the fast Li intercalation/deintercalation[43]. In contrast, the capacity retention is less than 80% for C-LNMO cycling at 10 C for only 200 cycles, and there is nearly no capacity output of C-LNMO when cycling at 20 C (Supplementary Fig. 13). Besides, the Co-LNMO showed an improvement of both coulombic efficiency and cycling stability at 1 C rate under 60 °C, with 80% capacity retention after 190 cycles, in contrast with 26% capacity retention for that in C-LNMO (Fig. 4i). Overall, these results reveal that the Co-LNMO enables significant improvements in structural stability and cyclability at a high rate/high temperature.

### Post-mortem analysis of Co-LNMO

Post-mortem analysis was used to elucidate the electrochemical performance difference between Co-LNMO and C-LNMO cathodes. The time-of-flight secondary ion mass spectroscopy (TOF-SIMS) snapshot images showed that the formation of glossy byproducts on the surface of cycled C-LNMO cathode, in contrast with little byproducts on cycled Co-LNMO (Fig. 5a). This can be ascribed to the less transition metal dissolution and side reaction between the electrode interface and the electrolyte after long-term cycling. There was a cathode electrolyte interface (CEI) layer formed on cycled Co-LNMO and C-LNMO, thus Ni signal was not detected in Ni 2p spectra at the surface (Fig. 5b). The structure vibration information of cycled cathodes was also not detected by Raman spectroscope due to the blocking CEI layer (Supplementary Fig. 14). As the Ar$^+$ sputtering, the Ni signal can be detected, and

the peak Ni $2p_{3/2}$ always keeps constant in the cycled Co-LNMO. In contrast, the peak Ni $2p_{3/2}$ shifts to higher binding energy in cycled C-LNMO after sputtering 5 min, and back to the lower binding energy position as the sputtering depth increases (Fig. 5b), indicating the high oxidation state of Ni near the surface and inconsistent valence distribution from the surface to the bulk in cycled C-LNMO. This is consistent with the Ni K-edge XANES results. There is no significant change in the XPS spectra of Mn 2p at different depths of both cycled samples (Supplementary Fig. 15), indicating the stable oxidation state of Mn.

XPS spectra of F 1s in both cycled cathode surfaces were used to reveal the CEI layer (Fig. 5c). There was a lower intensity of LiF/TMF$_x$ species in cycled Co-LNMO, indicating less transition metal dissolution and LiPF$_6$ decomposition[44,45]. The other peak assigned to C-F/P-F species is related to the fluorophosphate (Li$_x$PO$_y$F$_z$) composition, which is a favorable composition for restraining the interfacial side reaction[46]. The higher intensity of C-F/P-F species suggests fewer side reactions occurred during Co-LNMO cycling, in contrast to that in C-LNMO. Normalized TOF-SIMS depth profiling and 3D render of secondary ion fragments were used to analyze the specific composition in the CEI layer. More LiF$_2^-$ species formed by consuming Li from the active material or surface Li$_2$CO$_3$ were at the very surface of cycled Co-LNMO than that of cycled C-LNMO (Fig. 5d)[47], indicating more inorganic LiF species which benefits hindering the successive decomposition of the organic component in the electrolyte existed in the CEI layer of the cycled Co-LNMO. This can be also reflected in the 3D render of LiF$_2^-$ specie and concentration distribution (Fig. 5e). After sputtering for about 50 s, more LiF$_2^-$ species was detected in cycled C-LNMO, indicating more loss of active Li from the active material with more decomposition of the electrolyte[48]. It can be seen that the intensity of NiF$_3^-$ species was weaker in cycled Co-LNMO from the surface to the bulk than that in cycled C-LNMO (Fig. 5f), also shown from the 3D render of NiF$_3^-$ specie and concentration distribution and chemical imaging (Fig. 5g, h). It means less dissolution of Ni ions in the long-term cycling of Co-LNMO, which is closely related to the decrease of the amount of the highly oxidizing Ni ions. Overall, Co-LNMO has a reversible redox reaction during cycling, less structural degradation, and a stable CEI layer resulting in better electrochemical performance.

## Discussion

We have proposed a subtractive transformation strategy for D-NCM523 and D-LMO recycling and direct synthesis of a high-performance 5 V-class Co-LNMO that can meet cathode production in a self-sufficient mode without the addition of extra raw materials. Preferential extraction of equal proportions of Co and Ni from spent cathode materials led to a precise 0.1-mole ratio of Co left and substituted Ni in LNMO, which has been proven an optimal amount of Co for the Co-LNMO samples. The spinel Co-LNMO with a spherical secondary particle structure composed of octahedral primary particles has a more disordered cation ordering with improved conductivity, exhibiting better rate and cycling performance than C-LNMO for less transition metal dissolution and a more stable CEI layer formed during cycling. The improved bond strength, less structure degradation, and redox reversibility introduced by in-situ Co doping enable it to have high-rate (10 C/20 C, 1000 cycles, >80% capacity retention) and high-temperature (60 °C) performance. The D-NCM523 black mass can be recycled and transformed into high-performance Co-LNMO, showing potential for large-scale battery recycling. This strategy provides a concept of less input but more output in LIB recycling, opening a path for developing low-Co high-voltage cathode materials for a sustainable LIB industry.

## Methods

### Materials and recycling process

Spent NCM523 or LMO cathodes were obtained by disassembling of corresponding spent LIBs. The spent cathode materials were manually scraped from the cathodes and then dissolved in 4 M HCl (aq.) solution

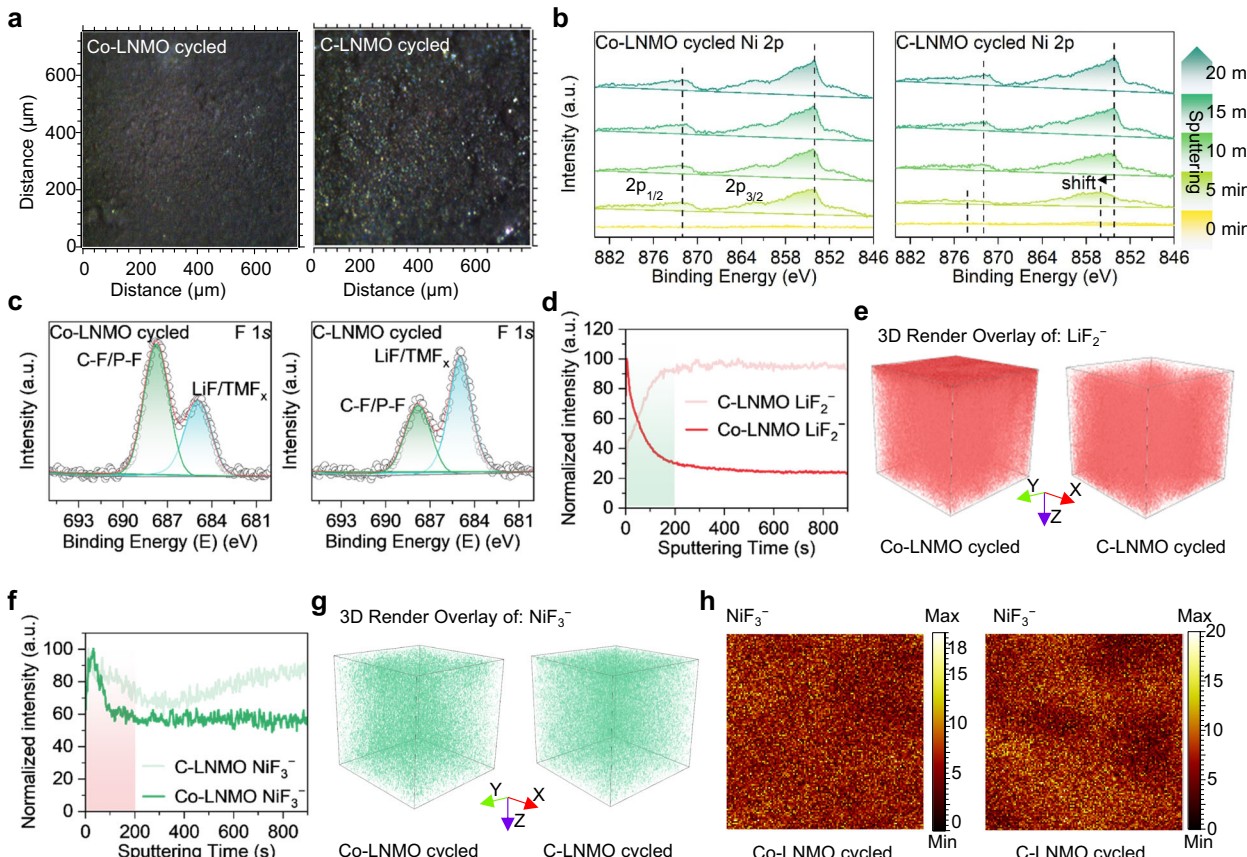

**Fig. 5 | Post-mortem analysis revealing the structure stability of Co-LNMO.**
**a** Snapshot images of the cathode surface in Co-LNMO and C-LNMO after the cycling test. **b** XPS spectra of Ni 2*p* at different depths of cycled Co-LNMO and C-LNMO. The acquisition of signal was collected every 5 min by Ar⁺ sputtering. **c** XPS spectra of F 1*s* at the surface of cycled Co-LNMO and C-LNMO. Normalized TOF-SIMS depth profiling of LiF₂⁻ (**d**) and NiF₃⁻ (**f**) Secondary ion fragment and 3D render of LiF₂⁻ (**e**) and NiF₃⁻ (**g**) specie and concentration distribution at the cycled Co-LNMO and C-LNMO cathodes. **h** TOF-SIMS chemical imaging of NiF₃⁻ specie at the cycled Co-LNMO and C-LNMO cathodes.

at 90 °C for 6 h with solid (S)-to-liquid (L) ratio (S/L = 5% w/v) and 800 rpm. The obtained dissolved solution (D-NCM523 and D-LMO) was filtered and washed with deionized (DI) water to remove the insoluble additives. The D-NCM523 black mass was supplied by Sinochem International Advanced Materials (Hebei) Corporation, and the dissolution process was as described above after the D-NCM523 black mass treated with 0.5 M NaOH (a.q.) for 10 min to dissolve possible aluminum impurities. The experiment was carried out with a maximum mass of 50 g black mass as raw material.

The separation of Ni and Co ions in the same valence state is difficult from the E-pH diagrams for the similar chemical behaviors of both elements[30]. Equal proportions of Ni and Co ions were preferentially extracted from a dissolved D-NCM523 solution. In this process, a certain amount of 0.1 M $(NH_4)_2C_2O_4$ (aq.) solution was added into a reactor in advance, and the D-NCM523 solution was pumped into the reactor (molar ratio of $Co^{2+}/C_2O_4^{2-}$ = 1: 1). And 0.3 M $NH_3 \cdot H_2O$ (aq.) was fed into the reactor at the same time. The reactor was placed into an oil bath at 55 °C for 1 h, and the reaction was controlled at pH = 2 and stirred at 400 rpm. The solution in the reactor was filtered and washed with DI water for the separation of the solid and liquid phases. The solid phase was then dried at 60 °C for 6 h to obtain the subtractive sediment for subsequent characterization. The liquid phase was collected and tested by ICP-OES to obtain element content for determining the amount of D-LMO solution addition.

A certain amount of dissolved D-LMO solution (based on the above ICP-OES result) was added into the above liquid phase solution to adjust the transition metal content in the solution for a mole ratio of

(Co + Ni): Mn = 0.5: 1.5, and the pH of the mixed solution was adjusted by 2 M $NH_3 \cdot H_2O$ (aq.) to get pH of 2 in a reactor. After that, all the transition metal ions in the solution were co-precipitated by 1 M $Na_2CO_3$ (aq.) solution, which was pumped into the continuously stirred reactor (molar ratio of $CO_3^{2-}$/transition metals = 1.1: 1). And 0.3 M $NH_3 \cdot H_2O$ (aq.) was used to adjust the pH of the solution. The reactor was placed into an oil bath at 55 °C, and the reaction was controlled at pH = 8.3 and stirred at 800 rpm overnight. After filtration and washing with DI water. The so obtained carbonate precursors were dried at 60 °C for 6 h and then heated at 850 °C for 5 h in the air to get oxide precursors. Then the obtained oxide precursors were thoroughly mixed with stoichiometric $Li_2CO_3$ and annealed at 850 °C for 10 h in air, the heating rate and cooling rate were 5 °C/min and 2 °C/min, respectively. A series of carbonate and oxide precursor samples and Co-LNMO samples with 0.05/0.1-mole ratios Co substitution of Ni or Mn were prepared based on the above process, which was used to investigate the Co substitution effects. The mole ratios of the Ni/Co/Mn in those samples were adjusted by using $NiCl_2$ (aq.) solution and dissolved D-LMO solution.

The residual solution was collected and tested by ICP-OES to obtain element content for lithium recovery. 1 M NaOH (aq.) solution was added to the residual solution to adjust the pH to 12, then the solution was heated to 90 °C by an oil bath. The saturated $Na_2CO_3$ solution was added to the solution (molar ratio of $CO_3^{2-}/Li^+$ = 1.2: 1) until getting the precipitate. The solid-liquid mixture was washed with boiled water, filtered, and dried at 80 °C for 12 h. The harvested product was labeled as a post-lithium precipitate.

## Preparation of electrode, battery assembly, and electrochemical measurements

The 80 wt.% of cathode materials (C-LNMO, Co-LNMO, and LNMOs with different proportions of Co substitution of Ni or Mn) with 10 wt.% of acetylene black and 10 wt.% of polyvinylidene fluoride (PVDF) were thoroughly mixed and dissolved in N-methyl-pyrrolidone (NMP) to form a cathode slurry. The slurry was continuously stirred overnight and cast on a carbon-coated aluminum foil. The foil was dried overnight at 80 °C under a vacuum to obtain the cathode electrode. The cathode electrode was cut into 12 mm round pieces with a mass loading of approximately 3-4 mg·cm$^{-2}$ for the assembly of CR2032 coin cell with a Li metal chip as the anode and Celgard2500 membrane as the separator. 1 M LiPF$_6$ in ethylene carbonate (EC): dimethyl carbonate (DMC): and diethyl carbonate (DEC) (1:1:1 vol.,) was used as the electrolyte. The C-LNMO was purchased from Haian Zhichuan Battery Material Co. LTD.

The galvanostatic charge-discharge measurements were carried out in the potential range of 3.0-4.95 V at 30 ± 1 °C using a Neware battery testing system (MIHW-200-160CH, Shenzhen, China). Rate capability was measured ranging from 1 C to 10 C (1 C = 146 mAh·g$^{-1}$). Cyclic voltammetry (CV) was performed using a Biologic VSP-3e electrochemistry workstation with a scan rate of 0.1 mV·s$^{-1}$. The GITT testing was performed using the LAND battery testing system with a 10 min constant current charge and a 30 min rest. The impedance changes at different voltages were collected by EIS measurement using an impedance analyzer (Biologic, VSP-3e). The charging-discharging process was conducted in a constant-current (CC) mode at 200 mA, and the electrochemical impedance spectroscopy at each voltage (4.6–4.9 V) was performed after a constant-voltage mode for 5 min. The high-temperature cycling performance was tested at 60 ± 1 °C.

## Material characterization

XRD patterns were obtained with a Rigaku Ultima IV powder diffractometer (Cu Kα radiation, 10–80° 2θ range). The relative molar ratios of Li/Ni/Co/Mn were determined by ICP-OES using a Perkin Elmer AvioTM 200 apparatus. The electronic conductivities of different Co-doped LNMOs were determined via the four-point probe testing method. The ionic conductivities of different Co-doped LNMOs were revealed by EIS measurement using an impedance analyzer (Biologic, VSP-3e). Taiwan Photon Source (TPS), National Synchrotron Radiation Research Center (NSRRC), provided 21A X-ray nano diffraction beamline which adopted a 4-bounce channel-cut Si (111) monochromator for Ni K-edge XANES and EXAFS measurements. Raman spectra were collected in the range from 100 cm$^{-1}$ ~ 750 cm$^{-1}$ using a Horiba Scientific LabRAM HR Evolution spectrometer. The top-view and cross-section images and morphology analysis were characterized by FIB-SEM (Helios G4 CX). TEM images and EDS maps were recorded with the FEI Talos F200x transmission electron microscope. XPS spectra at different depths were collected on a Thermo Scientific™ Escalab Xi$^+$ spectrometer with an Al K Alpha source gun. Multiple signals were collected after sputtering with an Ar$^+$ cluster ion source for each collection. The sputtering rate was 0.25 nm·s$^{-1}$ (Ta$_2$O$_5$ standard sample). For the post-mortem analysis, the cell was disassembled in the Ar-filled glovebox, and the cathode was washed with DMC, dried under vacuum, and sealed for XPS and TOF-SIMS measurements. All procedures were ensured to be carried out under air isolation conditions without any contamination. The chemical change of the cathode surface was analyzed using a TOF-SIMS spectrometer (ION-TOF GmbH TOF SIMS 5-100) with a Bi$^{3+}$ gun. A Cs$^+$ ion beam (1 KV 60 nA) maintaining a sputtering rate of 0.84 nm·s$^{-1}$ (GaN standard sample) was applied for sputtering the cycled cathodes for depth profiling. The area of typical sputtering samples was 200 × 200 μm.

## Reporting summary

Further information on research design is available in the Nature Portfolio Reporting Summary linked to this article.

## Data availability

The datasets generated during and/or analyzed during the current study are available from the corresponding author upon reasonable request. Source data are provided with this paper.

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

## Acknowledgements

G.Z. appreciates support from the Joint Funds of the National Natural Science Foundation of China (U21A20174), Guangdong Innovative and Entrepreneurial Research Team Program (2021ZT09L197), Shenzhen Science and Technology Program (KQTD20210811090112002), the Start-up Funds, Interdisciplinary Research and Innovation Fund of Tsinghua Shenzhen International Graduate School, and the Tsinghua Shenzhen International Graduate School-Shenzhen Pengrui Young Faculty Program of Shenzhen Pengrui Foundation (SZPR2023007). G.Z. and J.W. appreciate the support from Qinhe Energy Conservation and Environmental Protection Group Co., Ltd. (No. QHHB-20210405).

## Author contributions

J.M., J.W. and G.Z. conceived the project. J.M. conducted the experiments, fabricated the samples, performed electrochemical measurements, and analyzed the data with the assistance of K. J., G.J., H.J., Y.Z. and W.C. J.W., Z.L., H.-M.C. and G.Z. supervised the research and revised the manuscript. All authors discussed and contributed to the results. J.M. wrote the manuscript and J.M. and J.W. wrote the comments and revisions from all the authors.

## Competing interests

The authors declare no competing interests.
