## [Peer Review File · Nature Communications]

Subtractive transformation of cathode materials in spent Li-ion batteries to a low-cobalt 5 V-class cathode materialREVIEWER COMMENTS

Reviewer #1 (Remarks to the Author):

Introduction

Line 48: There are several new reviews and advances in recycling since 2020. This should be reflected.

Line 65: Same here, just one references is not really sufficient for the claim from 2019.

Results and discussion

Why is the commercial material running (electrochemically) so bad?

How many cells were compared for this study?

What is the difference of the proposed recycling compared to traditional hydrometallurgy?

The authors used acids and precipitation steps for example.

Reviewer #2 (Remarks to the Author):

The manuscript, "Subtractive transformation of cathode materials in spent Li-ion batteries to a low-cobalt 5 V-class cathode material," described a strategy for synthesizing Co-LNMO from spent cathode materials. The idea seems to be interesting. Some comments are below.

1. The subtractive method in this manuscript is based on hydrometallurgy, as 4M HCl(a.q.) was used to dissolve the cathode active materials. Will the regular hydro method be more proper in Figure 1 rather than direct recycling/upcycling, which should be more like an apple-to-apple comparison?

2. Some salts like NiCl₂ and Li₂CO₃ were added to tune with the ratios among different metal ions, which sounds additive rather than subtractive. I would suggest a workflow chart to clearly demonstrate the metal flow, input/output, and balance.

3. As the authors mentioned using cathode black mass as the feedstock, how were the carbon and binder removed? Will Al residual influence the process?

4. On Page 5, line 86, the authors mentioned that the method is scalable, which is critical for recycling. It would be helpful to provide the batch size (based on weight) in the method section.

5. The authors mentioned the uniformity of Co substitution at both the surface and the bulk in Fig 3j, which is not convincing. The Co 2p spectra showed an obvious decrease from surface to bulk, indicating the non-uniform distribution of Co. It would be necessary to have a reference sample, e.g., commercial NCM811, to demonstrate the Co uniformity.

6. Will this subtractive method be economically sound compared to regular hydrometallurgy?

Reviewer #3 (Remarks to the Author):

The manuscript discusses the utility of using degraded NMC as a spinel precursor.

1) the effort I believe focuses on dissolution of the NMC (lines 122-134) in acid and isolation of a mixed carbonate product by addition of I assume a carbonate precursor. Did the oxalate added convert to carbonate? (line 126). Seems like a modified hydroprocessing study.

2) the lithium was extracted as the carbonate - did it drag anything with it?

- 3) the figure 1 structures are not realistic, or representative of the materials involved.
- 4) the thermal conversion of delithiated layered solids to spinel is accepted. A recent 2021 Chem Comm paper led by Shi et al., converted $\text{Li}_x\text{MnNiO}_2$ to $\text{Li}(\text{MnNi})_2\text{O}_4$.
- 5) Co-doped 5V spinel is well known as a formulation to improve the properties of these high voltage but low-capacity materials.
- 6) the adjustment of solution ratio using the different solubility of the oxalates is useful but I'm not sure the extra cost of these analysis and separations is justifiable using TEA.

Point-by-point response to the reviewers' comments

General response:

We appreciate all the reviewers' professional comments and suggestions, with which we have improved the quality of our manuscript.

Reviewer #1 (Remarks to the Author):

Introduction

Line48: There are several new reviews and advances in recycling since 2020. This should be reflected.

Response:

Thanks for your suggestion, we have added recent progress in recycling to reflect the advances. Please see the highlighted areas in yellow.

“Sustainable production of rechargeable Li-ion batteries (LIBs) is important to meet the ever-growing market demand for electric vehicles and portable electronics^{1, 2}. However, limitations will arise in the context of increasing production requirements, one is that the raw materials required for today's LIBs chemistry are scarce and rarely sourced sustainably, and the other is poor sustainability in battery manufacturing and spent battery recycling³⁻⁵.”

Line 65: Same here, just one references is not really sufficient for the

claim from 2019.

Response:

Thanks very much for your kind suggestion, we have added recent progress in solving the problem claimed in line 65. We think the changes followed by your suggestion can make the claim more solid. Please see the highlighted area in yellow.

“However, the problem of transition metal dissolution and inferior high-temperature performance of LNMO hindered its application ¹⁴⁻¹⁶”

Results and discussion

Why is the commercial material running (electrochemically) so bad?

Response:

Thanks for your comment. Compared to the reported work (*Journal of Materials Chemistry A* 2023 11(14): 7670-7678 and *ACS Omega* 2019 4(1): 185-194), the electrochemical performance of pure LNMO (350 cycles under 1 C rate, ~ 60% capacity retention and 50 cycles under 55°C, < 50% capacity retention) is similar to or even worse than those of LNMO shown in our manuscript. During cycling at 1 C rate, the commercial LNMO showed ~74% capacity retention after 400 cycles, which is at the same level of reference and is slightly lower than that of Co-LNMO (>80% capacity retention). The biggest differences in the electrochemical performance of both cathode materials were in the high-rate and high-temperature cycling. The electronic and ionic conductivity of commercial material with pure LNMO spinel structure is relatively poor compared with the doped samples (reflected in Fig. 2h, i). The lower conductivities of

the materials, the more sluggish electronic and ionic diffusion ability. It shows poor compatibility between the material and the electrolyte which leads to bad interface stability (shown in Fig. 4 and Fig. 5). When cycling at a 10 C rate, the commercial material shows significant capacity decay due to the lower conductivities and interface stability. When cycling at high temperature, there were more concerns about transition metal ion dissolution, but the unstable interphase could not prevent the probability of dissolution, leading to somewhat bad electrochemical performance. In addition, the grain diameter of the primary particle of commercial material is smaller, there was more contact between the particle and electrolyte forming an interphase accompanied by the decomposition of the electrolyte. The electrolyte we used is a common formula with EC : DMC : DEC (1:1:1 vol.) solvents, which lacks high-voltage additives, making the commercial material with an inferior interface when cycling at high voltage. In contrast, the resynthesized material has a more stable crystal structure and is easy to form a stable interface due to Co doping, it shows better electrochemical performance.

How many cells were compared for this study?

Response:

Thanks for your comment. During the study process, we repeated every experiment to obtain the samples. Three to six cells using the same batch of sample were tested simultaneously for comparison, which we think is reasonable and convincing for the results.

What is the difference of the proposed recycling compared to traditional hydrometallurgy?

The authors used acids and precipitation steps for example.

Response:

Thanks for your comment. The difference is that we designed the proposed recycling for precise control of the proportional composition of the elements to get the precursor with an ideal stoichiometric ratio, which is difficult due to the similar properties and precipitation behavior of Ni, Co, and Mn. After processing parameters optimization, the proposed recycling can save steps from the spent to the product. The precipitation step in traditional hydrometallurgy is for obtaining a single-component transition metal salt precursor, which involves a multi-step separation process for every single element, and the overall process is longer. The precipitation step in the proposed recycling route is for the direct production of cathode precursor, which has no cumbersome separation steps for every single-component salt and the overall process is shorter (see **Fig. R1**)

Fig. R1 Material flow analysis of the traditional hydrometallurgical recycling process and the proposed recycling strategy.

The effort we focus on is the preferential isolation of partial ions by oxalate, leaving residual ions in the solution proportional, which can be transformed into a precursor with a definite stoichiometric ratio by the carbonate co-precipitation method. Dissolution using acids is just a step to get the ionic state of elements, which is not paying close attention to. Some green and functional solvents, e.g. deep-eutectic solvents, can be used for dissolution (Mai K. Tran et al. *Nature Energy*, 2019, 4(4): 339-345), but this is not focused on in this manuscript.

Reviewer #2 (Remarks to the Author):

The manuscript, "Subtractive transformation of cathode materials in spent Li-ion batteries to a low-cobalt 5 V-class cathode material," described a strategy for synthesizing Co-LNMO from spent cathode materials. The idea seems to be interesting. Some comments are below.

Response:

We appreciate the reviewer's recognition of our work and also thank the instructive suggestion for us to look forward to an outlook on the manuscript.

1. The subtractive method in this manuscript is based on hydrometallurgy, as 4M HCl(a.q.) was used to dissolve the cathode active materials. Will the regular hydro method be more proper in Figure 1 rather than direct recycling/upcycling, which should be more like an apple-to-apple comparison?

Response:

Thanks for your kind suggestion, it seems similar between the subtractive method and the traditional hydrometallurgy method due to the use of acid for dissolution. However, using acids is just a step to get the ionic state of the element. The key of this work is on the preferential isolation of partial ions by oxalate, leaving residual ions in the solution proportional, which can be transformed into a cathode precursor with a definite stoichiometric ratio. The product of the regular hydro method is a single-component metal salt precursor, e.g. NiCO_3 , MnCO_3 , CoCO_3 , and, Li_2CO_3 , the products of direct

recycling/upcycling toward cathode, which are comparable with the subtractive method. In addition, value-added products may be obtained by both methods, which have better properties and performance than ordinary materials. The proposed recycling processes cost less, subtracting specific components in the waste beforehand can facilitate the recycling process to a self-sufficient mode of sustainable production, in sharp contrast to adding extra raw materials for the direct recycling or upcycling of the spent to the new. From these points of view, we consider comparing the two methods together to emphasize the differences between additional investment for recycling and self-sufficient mode for recycling.

Your suggestion is also reasonable, so we also added the comparison between the regular hydro method and the subtractive method in the Supplementary Information (see Fig. S1)

Fig. S1 Comparison of the subtractive strategy and the regular hydrometallurgical strategy

2. Some salts like NiCl_2 and Li_2CO_3 were added to tune with the ratios among different metal ions, which sounds additive rather than subtractive. I would

suggest a workflow chart to clearly demonstrate the metal flow, input/output, and balance.

Response:

Thanks for your comment and kind suggestion, NiCl₂ was used to adjust the Ni mole ratio in the series of Co-doped samples for investigation of the Co substitution effects. But the final product (LiCo_{0.1}Ni_{0.4}Mn_{1.5}O₄) with 0.1-mole ratio of Co substitution of Ni does not require NiCl₂ for the mole ratio adjustment due to a nearly equal mole ratio of Ni and Co was preferentially extracted, leaving residual with the relative Ni/Co/Mn molar ratios of 0.4, 0.1, 1.5. Li₂CO₃ was recovered from the residual solution after cathode precursor harvesting, no additional Li₂CO₃ was input in the recycling process. A workflow chart to demonstrate the metal flow, input/output, and balance was presented as follows (**Fig. R2**). In addition to the use of conventional precipitators and pH-regulating ammonia, the subtractive process does not include any additional input of metal salts.

Fig. R2 A workflow chart of this subtractive strategy

3. As the authors mentioned using cathode black mass as the feedstock, how were the carbon and binder removed? Will Al residual influence the process?

Response:

Thanks for your comment. The cathode black mas was supplied by the Sinochem International Advanced Materials Corporation which was harvested after the fine disassembly and separation processes. There were few coarse materials, and the PVDF/carbon black and cathode active materials were more likely in the state of individual components, in stark contrast to those in a cake. Carbon black has a porous and multi-particle structure, and the upward buoyancy formed by the contact between the particularity and the air makes it float in the water. It has a large specific surface area, the surface is rich in oxygen, nitrogen, and other functional groups, these functional groups can form strong adsorption with water molecules, further increasing the possibility of carbon black floating on the water. With the dissolution of cathode active materials, the adhesion between PVDF/carbon black and cathode active materials becomes weaker, making the liberation of individual components to be separated. Since PVDF and carbon black are hydrophobic (*Kuo et al., Desalination, 2008, 233, 40-47; Wang, et al., Electrochim. Acta, 2006, 51, 4909-4915*), cathode active materials are hydrophilic (*He et al., J. Cleaner Prod. 2017, 143, 319-325*), the PVDF/carbon black were mostly in the upper of the aqueous solution during the agitation and the gas production with dissolution process, which can be removed by centrifugation and filtration. There was little Al residual in the black mass for fine disassembly and separation processes to delaminate the cathode black mass from current collectors by the company. The black mass was treated with 0.5 M NaOH (a.q.) for 10 min to dissolve any aluminum impurities to eliminate the influence of possible

aluminum impurities. We have supplemented this in the method part.

4. On Page 5, line 86, the authors mentioned that the method is scalable, which is critical for recycling. It would be helpful to provide the batch size (based on weight) in the method section.

Response:

Thanks for your kind suggestion, we have verified the case of using 50 g black mass as raw material and recycling by the proposed subtractive method. Unfortunately, we have only been able to verify it on this scale, because the laboratory lacks a reaction vessel with a larger volume. Precisely, this method has the potential to scale up. Your reminder is very professional, and we have made corresponding supplements in the method part.

5. The authors mentioned the uniformity of Co substitution at both the surface and the bulk in Fig 3j, which is not convincing. The Co 2p spectra showed an obvious decrease from surface to bulk, indicating the non-uniform distribution of Co. It would be necessary to have a reference sample, e.g., commercial NCM811, to demonstrate the Co uniformity.

Response:

Thanks for your comment, we have checked the result shown in Fig. 3j in the previous manuscript. The Co 2p spectra showed a decrease from surface to bulk, which was also shown in the Mn 2p and Ni 2p spectra. It seems like the non-uniform distribution may be affected by the detected area where the information was collected with XPS. With

the etching process, the amounts of materials in the detected area may vary due to the secondary-particle structure with an inner void, so the overall intensities of elements may be lowered from the surface to the bulk, as shown in **Fig 3 j-l** from the previous manuscript.

Fig. 3 j-l, XPS spectra of different elements at different depths in Co-LNMO: the Co 2p (**j**), the Mn 2p (**k**), and the Ni 2p (**l**).

To alleviate the affection from the detection area, we have retested the XPS spectra of different elements at different depths in Co-LNMO, and the result shows nearly the same intensities from the surface to the bulk, which can be seen from the Co 2p, Mn 2p, Ni 2p spectra, indicating the uniform distribution of these elements from the surface to the bulk.

Fig. 3 j-l, XPS spectra of different elements at different depths in Co-LNMO: the Co 2p (**j**), the Mn 2p (**k**), and the Ni 2p (**l**).

We have also followed your suggestion to have a reference sample, the XPS spectra of

different elements at different depths in commercial NCM811 (**Fig. R3**) were collected to demonstrate the Co uniformity from the surface to the bulk.

Fig. R3 XPS spectra of different elements at different depths in NCM811

6. Will this subtractive method be economically sound compared to regular hydrometallurgy?

Response:

Thanks for your comment, we have made a specific workflow of the subtractive method and the regular hydrometallurgical route (the hydro), see **Fig. R4** as follows.

Fig. R4 Material flow analysis of the regular hydrometallurgy and the subtractive method

The processing steps involved in this strategy are fewer in contrast to the multi-step component separation processes in the hydro. Suppose that 1 ton of ternary NCM523

spent batteries are treated, the separation efficiencies of each element is 100% and the cost for obtaining product is simplified in the hydro. The material cost of this strategy is lower than that of the hydro (~409 \$). And the income of the hydro is lower than that in this strategy, delivering the profit (~1,510 \$) of the hydro, which is lower than the subtractive method (~2,676 \$) (**Fig. R5**). The specific price of materials and some other costs are presented in **Table R1-2**.

Fig. R5 Economic analysis of the conventional hydrometallurgical recycling process and this subtractive method

Table R1 The specific price of different materials involved in TEA analysis

The specific price of different materials involved in TEA analysis (The prices are referenced from China Metals Network (www.cbcie.com))	
Materials	Price (\$/t)
Spent NCM LIBs	3318.9
Spent LMO LIBs	1341.99
HCl	40.404
Na ₂ CO ₃	432.9
NH ₄ C ₂ O ₄	1760.46
NH ₂ H ₂ O	209.235
Li ₂ CO ₃	30548.31
NiO	23556.975
CoO	35714.25
LNMO	14430
Water	0.21645
NaOH	360.75
NH ₄ HCO ₃	194.805
MnCO ₃	779.22

Table R2 Some other costs during practical LIBs recycling

Some other cost during practical LIBs recycling	
Items	Cost (\$/t-cell)
Direct labor	150
Depreciation	1500
Variable overhead	450
Generals, sales, administration	640
Research and development	500
all	3240

These cost estimates were primarily derived from the EverBatt2020 database

Reviewer #3 (Remarks to the Author):

The manuscript discusses the utility of using degraded NMC as a spinel precursor.

1) the effort I believe focuses on dissolution of the NMC (lines 122-134) in acid and isolation of a mixed carbonate product by addition of I assume a carbonate precursor. Did the oxalate added convert to carbonate? (line 126). Seems like a modified hydroprocessing study.

Response:

Thanks for your comment, the effort we focus on the preferential isolation of partial ions by oxalate, leaving residual ions in the solution proportional, which can be transformed into a precursor with a definite stoichiometric ratio by the carbonate co-precipitation method. Dissolution is just a step to get the ionic state of elements, which is not paying close attention to. Some new solvents, e.g. deep-eutectic solvents, were researched for dissolution (Mai K. Tran et al. *Nature Energy*, 2019, 4(4): 339-345), but this is not focused on in this manuscript. Maybe some parts were not clearly described in the previous version of the manuscript, so we have made some changes. Please see the revised part in lines 125-133

“To obtain an ideal stoichiometric ratio of each element for the direct preparation of the precursor, (NH₄)₂C₂O₄ solution was used for preferential subtraction of both Ni²⁺ and Co²⁺ ions from the dissolved D-NCM523 solution to form oxalate precipitate by

making full use of the similar chemical behavior in the aqueous solution of both elements when the pH is low in acidic condition^{26, 27}. The residual ions remain in the solution due to complexation with NH_4^+ (Bottom right in Fig. 1). A carbonate $Ni_xCo_yMn_{1.5}(CO_3)_2$ ($x + y = 0.5$) precursor was obtained by a simple co-precipitation method with the addition of Na_2CO_3 after using the dissolved D-LMO solution for desired Mn stoichiometric ratio regulation”

In line 126, the oxalate added was for the preferentially extract of partial Ni and Co ions in the solution to form oxalate precipitate, it did not convert to carbonate. The carbonate $Ni_xCo_yMn_{1.5}(CO_3)_2$ ($x + y = 0.5$) precursor with a specific stoichiometric ratio was obtained by a carbonate (with the addition of Na_2CO_3) co-precipitation process. Although some steps like dissolution and precipitation seem like in the hydroprocessing, the core of the proposed strategy is not the dissolution or efficient separation of each component. The easier conversion of the spent (one of mixed) to the new in a controllable way is what we focus on. The amount of oxalate was optimized to enable controllable extraction of some Co/Ni preferentially, and the molar content of oxalate was the same as that of Co/Ni that we wanted to preferentially extract. After the pre-extraction process, the amount of transition metal ions in the residual solution is proportional, leading to the production of Co-doped LNMO precursor for 5 V-class cathode material preparation. This work attempts to simplify the processes of composition regulation, multiple separation, and purification and builds a bridge between the spent cathode material and a high-performance cathode material. It is different from the regular hydroprocessing study.

2) the lithium was extracted as the carbonate - did it drag anything with it?

Response:

Thanks for your comment, the solution obtained after the co-precipitation process has almost no residual transition metal ions, which can be verified by the high reuse rate (>99.9%) of every transition metal element (**Fig. 2c**).

Fig. 2c Element concentration of the pristine D-NCM523 solution and its solution after the co-precipitation process, and the corresponding reuse rate of every metal element.

In addition, from the XRD and ICP-OES results, we could see that the lithium extracted as the carbonate has a monoclinic crystal structure with the *I2/a* space group, corresponding to the XRD pattern of Li₂CO₃ (**Fig. R6**). The element concentration of the carbonate is shown in **Table R3**, which suggests a very low content of impurities in the carbonate.

Fig. R6 XRD pattern of the post-lithium precipitate

Table R3 ICP-OES result of the post-lithium precipitate

Element	Concentration (mg/kg)	Mass ratio %
Co	171.82	0.0958%
Li	178827.60	99.6782%
Mn	255.21	0.1423%
Ni	150.31	0.0838%

3) the figure 1 structures are not realistic, or representative of the materials involved.

Response:

Thanks for your suggestion, there is indeed a structure that may not represent the material involved, which we have revised following your suggestion (see revised **Fig.**

1). We think the revised can be more realistic and representative.

Fig. 1 Schematic of the additive strategy and subtractive strategy

4) the thermal conversion of delithiated layered solids to spinel is accepted. A recent 2021 Chem Comm paper led by Shi et al., converted $\text{Li}_x\text{MnNiO}_2$ to

Li(MnNi)2O4.

Response:

Thanks for your comment. We have systematically read the paper you have mentioned. Shi et al reported a new, polymorphic form of $\text{LiMn}_{0.5}\text{Ni}_{0.5}\text{O}_2$, which has a partially-disordered rock salt structure with a predominant lithiated-spinel-like character. In this manuscript, this cathode material, denoted $\text{LT-LiMn}_{0.5}\text{Ni}_{0.5}\text{O}_2$ (or $\text{LT-Li}_2\text{MnNiO}_4$ in lithiated-spinel notation), is different from the layered, ‘high-temperature’ $\text{LiMn}_{0.5}\text{Ni}_{0.5}\text{O}_2$ polymorph. It was synthesized by the solid-state reaction of Li_2CO_3 and $\text{Mn}_{0.5}\text{Ni}_{0.5}(\text{OH})_2$ precursors in the air at 400°C , which shows a two-phase character by Rietveld refinement. The average degree of disorder ($\sim 16\%$) in such $\text{LT-LiMn}_{0.5}\text{Ni}_{0.5}\text{O}_2$ model structures ($\sim 16\%$ of the Li ions on the 16c sites were exchanged with Mn/Ni ions on the 16d sites) lies between that observed in layered $\text{HT-LiMn}_{0.5}\text{Ni}_{0.5}\text{O}_2$ ($\sim 11\%$) and the cation distribution in an ideal, ordered lithiated-spinel structure, such as $\text{Li}_2\text{Mn}_2\text{O}_4$, in which 25% of the transition metal ions reside in 16d sites in the lithium-rich layers, and 25% of the lithium ions reside in 16c sites in the transition-metal-rich layers. It shows two main structure change driven by the electrochemical processes, the one that lithium extraction from octahedral sites and nickel oxidation in the delithiated spinel and layered regions of the electrode structure, respectively, and the other that lithium extraction from tetrahedral sites and nickel oxidation in the spinel and layered domains. The Rietveld refinement result shows that a delithiated $\text{LT-Li}_{1-x}\text{Mn}_{0.5}\text{Ni}_{0.5}\text{O}_2$ electrode after charging to 4.2 V is consistent with a spinel-like structure in which Li occupies tetrahedral sites. It involves the gradual migration of cations from one layer to the next

driven by the electrochemical reaction, which is different from the thermal conversion. In our manuscript, we studied a typical disordered spinel structure with an Fd-3m space group, transition metal ions are randomly distributed in the 16d site of the octahedron, and lithium ions occupy the 8a site of the tetrahedron. The lithium extraction process involves the move of lithium ions from the tetrahedral 8a site to an empty octahedral 16c site. Although the structure of materials re-synthesized is different from that in the paper, the concept of manipulating the order and disorder degree of cations in the structure of materials is similar. We believe that this paper has reference value for deepening the understanding of the property changes caused by the cation distribution of structure, so we also refer to order and disorder content in the corresponding part of the paper and quote it in the manuscript (B. Shi, et al. *Chemical Communications* 2021, 57(84): 11009-11012) Thank you again for your reminder and suggestion

5) Co-doped 5V spinel is well known as a formulation to improve the properties of these high voltage but low-capacity materials.

Response:

Thanks for your comment. Although the Co substitution in spinel LNMO reduces a certain capacity due to the lowering of redox ions that can contribute capacity. The Co-doped samples just show $\sim 5 \text{ mAh}\cdot\text{g}^{-1}$ reduction compared to LNMO (Fig. 2g)

Fig. 2g Charge and discharge curves of a series of LNMO samples with Co substitution (0.05/0.1-mole ratios) of Ni or Mn.

However, these samples showed improved structural stability and redox reversibility with improved rate and cyclic stability, especially when cycling at a high rate and high temperature (Fig. 4c, h, and i). Compared to LNMO, the Co-LNMO showed higher capacity at 2 C, and $\sim 10 \text{ mAh}\cdot\text{g}^{-1}$ and $\sim 20 \text{ mAh}\cdot\text{g}^{-1}$ improvement at 5 C and 10 C. The application of this material can be in fast-charging conditions and maintain better long-term stability under this condition.

Fig. 4c Rate capability measurements of Co-LNMO and C-LNMO in a constant-current (CC) mode, $1 \text{ C} = 146 \text{ mAh}\cdot\text{g}^{-1}$. **h**, Capacity retention of Co-LNMO and C-LNMO during cycling at a 10 C rate at 30°C for 2000 cycles. **i**, Cycling performance of Co-LNMO and C-LNMO at 60°C.

Co has been used as a doping element for improving the properties of different cathode materials (layered, spinel, and olivine). But more attention is paid to the effect of introducing another element from the intrinsic structure of the material. Here, we are concerned with removing/lowering the amount of one element from the intrinsic structure of the material so that the remaining may work better to form a new structure and deliver good performance. What we want to emphasize more is how to reduce the proportion of the one/two in a controllable way to easily get the ideal composition of the material and to explore the reasons for property promotion during the recycling process. By the adjustment of solution ratio using the different solubility of the oxalate, a specific amount of the rare and high-cost element is successfully lowered, leaving a

proportional precursor with Co doping naturally. In addition, this strategy builds a bridge from the spent to the product. We have verified the sample with the 0.1-mole ratios Co substitution of Ni ($\text{Co}_{0.1}\text{Ni}_{0.4}$) showed stronger TM-O bonds and Li^+ diffusion capability with improved structural stability and redox reversibility compared with other samples with different Co doping ratios. This may provide some guidance on a direct path from the spent to the product with a specific composition during the recycling and production process.

6) the adjustment of solution ratio using the different solubility of the oxalates is useful but I'm not sure the extra cost of these analysis and separations is justifiable using TEA.

Response:

Thanks for your comment, we have taken the TEA of this strategy and the conventional hydrometallurgical strategy (the hydro), the materials flow was seen in **Fig. R7**.

Fig. R7 Material flow analysis of the conventional hydrometallurgical recycling process and this strategy

The processing steps involved in this strategy are fewer in contrast to the multi-step component separation processes in the hydro. Suppose that 1 ton of ternary NCM523 spent batteries are treated, the separation efficiencies of each element is 100% and the cost for obtaining product is simplified in the hydro. The material cost of this strategy is lower than that of the hydro (~409 \$). And the income of the hydro is lower than that in this strategy, delivering the profit (~1,510 \$) of the hydro, which is lower than our strategy (~2,676 \$) (**Fig. R8**). The specific price of materials and some other costs are presented in **Table R1-2**.

Fig. R8 Economic analysis of the conventional hydrometallurgical recycling process and this strategy.

Table R1 The specific price of different materials involved in TEA analysis

The specific price of different materials involved in TEA analysis (The prices are referenced from China Metals Network (www.cbcie.com))	
Materials	Price (\$/t)
Spent NCM LIBs	3318.9
Spent LMO LIBs	1341.99
HCl	40.404
Na ₂ CO ₃	432.9
NH ₄ C ₂ O ₄	1760.46
NH ₃ H ₂ O	209.235
Li ₂ CO ₃	30548.31
NiO	23556.975
CoO	35714.25
LNMO	14430
Water	0.21645
NaOH	360.75
NH ₄ HCO ₃	194.805
MnCO ₃	779.22

Table R2 Some other costs during practical LIBs recycling

Some other cost during practical LIBs recycling	
Items	Cost (\$/t-cell)
Direct labor	150
Depreciation	1500
Variable overhead	450
Generals, sales, administration	640
Research and development	500
all	3240

These cost estimates were primarily derived from the EverBatt2020 database

REVIEWER COMMENTS

Reviewer #2 (Remarks to the Author):

I appreciated authors' effort during the revision—a follow-up comment for the authors to consider.

Fig. S1, Fig. R2, and Fig. R1 seem different, but are all referred to the subtractive method? I may maintain my original point that the subtractive method is quite like hydrometallurgy. The subtractive concept will be sounder if some elements can be subtracted without breaking down the whole cathode materials.

Reviewer #3 (Remarks to the Author):

1) the authors have addressed several of my comments but in the end I think its a selective isolation of Co from a mixture then reconsolidating the mixture (w less cobalt) to a spinel system.

2) unless at higher temperatures, LNMO should cycle well, especially the pristine version. Why might the capacity fade so fast?

Point-by-point response to the reviewers' comments

General response:

We appreciate all the reviewers' professional comments and suggestions, with which we have improved the quality of our manuscript.

Reviewer #2 (Remarks to the Author):

I appreciated authors' effort during the revision—a follow-up comment for the authors to consider.

Fig. S1, Fig. R2, and Fig. R1 seem different, but are all referred to the subtractive method? I may maintain my original point that the subtractive method is quite like hydrometallurgy. The subtractive concept will be sounder if some elements can be subtracted without breaking down the whole cathode materials.

Response:

Thanks for your comment, the core content of **Fig. S1**, **Fig. R1**, and **Fig. R2** in the last version of the manuscript and response letter is the same, only **Fig. S1** is a brief description of the simplicity of this subtractive method, and no specific material flow. **Fig. R1** and **Fig. R2** are more detailed comparisons, of each material used to the exact amount of the final product comparison. We are sorry that we have overlooked the potential ambiguity of several description forms, and we have changed and unified them,

please see the **Supplementary Fig. 1**, all appear in this form for comparison.

Fig. 1 The comparison of the subtractive strategy and the regular hydrometallurgy strategy.

Your comments and suggestions let us continue to dig deeper into our proposed method. It is ideal if some elements can be subtracted without breaking down the whole cathode materials, we are also looking forward to seeking the possible method. We tried the ion exchange method and selective adsorption method like using a metal-organic framework (MOF) membrane for ion screening. However, the metal oxide with layer structure and stable chemical bonds in cathode materials makes it difficult to subtract proportionate elements. The breaking of the chemical bond (e.g. TM-O bond) will lead to the distortion or even collapse of the crystal structure. It is impossible to achieve arbitrarily proportional chemical bond break without structural change. Maybe some method in the delithiation process can give inspiration for selective subtraction without breaking down the material structure. It is also difficult to directly prepare a structure with a chemical composition ratio like LNMO. In this work, we select ammonium

oxalate as a subtractive agent for the preferentially subtraction of some elements by regulation of pH and feeding speed, which can not only solve the precise separation and subtraction of specific elements but also directly prepare the precursor with the chemical composition ratio of LNMO. The idea of Less is More in recycling can also inspire more research, thinking about how we implement selective subtraction considering scarcity and the high-value nature of some elements, e.g., Co, from waste to the next generation of high-performance cathode material products. The dissolution is just a step to make it easy for the selective extraction of some elements. Though it is commonly used in hydrometallurgy, the subsequent selective subtraction and recycling route from waste to wealth have never been reported.

To better compare the differences in methods, we researched some literature about the separation and recovery of transition metals (Ni, Co, Mn, Al, etc.) from spent ternary cathode by the regular hydrometallurgy methods (**Supplementary Table 1**). Some works studied the leaching and separation of mixed metals but required many organic extractants for high-efficiency recovery¹⁻³. For example, Li et al.¹ use a combination of precipitation and solvent extraction for the separation and recovery of Ni, Co, Mn, and Li. Ni is extracted as Ni-(C₄H₈N₂O₂)₂, and the transition metal solution is mixed bis-2-ethylhexylphosphoric acid (P204), bis(2,4,4-trimethylpentyl) phosphinic acid (C272) according to a certain oil-aqueous ratio and extracted in steps of Mn²⁺ and Co²⁺ using reciprocating horizontal extraction oscillator. After that, Mn and Co are recovered in the form of precipitates, and the complex-precipitation of Ni-(C₄H₈N₂O₂)₂ can be dissolved in sulfuric acid and recrystallization to recycle DMG and NiSO₄ respectively.

Although the recovery rate of Ni and Mn is 96.84% and 92.65%, respectively, the recovery rate of Co is only 81.46%. The process steps are relatively cumbersome. Ilyas et al. ⁴ developed the selective separation of Co versus Ni using a green ionic liquid (IL), trihexyl(tetradecyl)phosphonium bis-2,4,4-(trimethylpentyl)phosphinate (>\$18/g), where more than 99% of Co was selectively extracted. However the use of KMnO₄ for Mn separation and the high cost of that IL hinders large-scale application of this recycling route. Further processing is required to obtain the product from which Co and Ni. Joulié et al.⁵ also improved the Co recovery rate and did not require organic reagents. but the oxidative precipitation efficiency changes due to the use of chemically unstable NaClO, which shifts the balance reaction to Co oxidation and separation ^{5,6}. Some studies demonstrate a hydrometallurgy process to be capable of selective leaching of partial elements from spent materials, which do not study the subsequent separations of single elements due to the separation difficulties. ^{7,8}

Table 1 Comparison of regular hydrometallurgical recycling methods from the reported works

Cathode material	Reagent	Product	Recovery rate
NCM ¹	H ₂ SO ₄ and H ₂ O ₂ , (C ₄ H ₈ N ₂ O ₂), P204 and C272	DMG Ni-(C ₄ H ₈ N ₂ O ₂) ₂ , MnO ₂ , CoC ₂ O ₄ and Li ₂ CO ₃	Ni (96.84%), Co (81.46%), and Mn (92.65%)
NCA ²	H ₂ SO ₄ , H ₂ O ₂ ethylhexyl) phosphoric acid,	D2EHPA-Bis(2- Co ₃ O ₄ (purity 83%), CoC ₂ O ₄	Co (80%~85%), Ni (90%) Li

	Cyanex272®-Bis	2,4,4-	(purity 96%), NiO (72%).
	trimethylpentyl phosphinic acid	(purity 89%), and	
		Li ₂ CO ₃ (purity	
		99.8%)	
NCM ³	Reduction roasting with starch,	No	Co (90.9%), Li
	H ₂ SO ₄ , Mextral 984H (5-		(82%), Ni and Cu
	nonylsalicylaldoxime and 2-		(98%)
	hydroxy-5-nonylacetophenone		
	oxime),		
NCM ⁴	KMnO ₄ ,	IL	split-phosphinate extraction
	(trihexyltetradecylphosphoniu	molecule (as CoA ₂)	equilibria of Co ²⁺
	m	bis(2,4,4-	and stabilization of with ionic liquid
	trimethylpentyl)phosphinate),	phosphonium	(>99%)
	aromatic solvent C10, NaOH	molecule	by
		complexing with the	
		chloride ions of	
		aqueous solution (as	
		R ₄ P ⁺ Cl ⁻) of the used	
		ionic liquid (i.e.,	
		R ₄ P ⁺ A ⁻).	
NCA ⁵	HCl, NaClO, NaOH	Co ₂ O ₃ ·3H ₂ O,	Co (90.25%) and
		Ni(OH) ₂	Ni (96.36%), Li

			(>80%)
NCM ⁶	HCl, hypochlorite solution (NaClO), Na ₂ CO ₃	Mixture of Mn (95%) and Co manganese oxides (90%) (MnO ₂ , Mn ₃ O ₄ , Na _{0.55} Mn ₂ O ₄ ·1.5 H ₂ O, CoCO ₃ , and Li ₂ CO ₃	
NCM ⁷	NH ₃ , (NH ₄) ₂ SO ₄ , Na ₂ SO ₃	(NH ₄) ₂ Mn(SO ₃) ₂ · H ₂ O, Li ₂ SO ₄ , amine complexes	NA, Leaching rate: Ni (94.8%), Co (88.4%), Mn (6.34%), and Li (96.7%)
NCM ⁸	(NH ₄) ₂ SO ₄ , (NH ₄) ₂ SO ₃	Li ₂ SO ₄ , (NH ₄) ₂ Co(SO ₄) ₂ · H ₂ O, (NH ₄) ₂ Mn(SO ₃) ₂ · H ₂ O and (NH ₄) ₂ Mn(SO ₄) ₂ · 6H ₂ O	NA, Leaching rate: Ni (98%), Co (81%), Mn (92%), and Li (98%)

The subtractive method is for precise control of the proportional composition of the elements to get the precursor with an ideal stoichiometric ratio, which is difficult due to the similar properties and precipitation behavior of Ni, Co, and Mn. After

processing parameters optimization, the proposed recycling can save steps from the spent to the product, and reduce the use of extractants. The solution obtained after the co-precipitation process has almost no residual transition metal ions, which can be verified by the high reuse rate (>99.9%) of every transition metal element (**Fig. 2c**).

Fig. 2c Element concentration of the pristine D-NCM523 solution and its solution after the co-precipitation process, and the corresponding reuse rate of every metal element.

In regular hydrometallurgy methods, they are for obtaining a single-component transition metal salt. A variety of precipitants/extractants even some hazardous and costly ones are required which involves a multi-step separation process for every single element, and the overall process is longer, as **Supplementary Fig. 1** shows. Suppose that 1 ton of ternary NCM523 spent batteries are treated and the regular hydrometallurgy strategy uses the common and inexpensive inorganic extractants, that the recovery rate is 100%, and that the recovered products are all transformed into commercial products without further purification. The processing steps involved in the subtractive strategy are fewer in contrast to the multi-step separation processes in the regular hydrometallurgy strategy. The material cost of the hydrometallurgy strategy is higher (~409 \$) than that of the subtractive strategy. And the profit (~1,510 \$) of the hydrometallurgy strategy is lower than this subtractive strategy (~2,676 \$) (**Fig. X1**).

Fig. X1 Economic analysis of the conventional hydrometallurgical recycling process and this strategy

Based on the above analysis, we think that the proposed subtractive method and the concept are innovative and significant. We are going to study more green and straightforward subtraction methods in the following research. Thank you again for your suggestions and the ongoing discussion, which has played an important role in this paper and the subsequent research.

Reviewer #3 (Remarks to the Author):

1) the authors have addressed several of my comments but in the end I think its a selective isolation of Co from a mixture then reconsolidating the mixture (w less cobalt) to a spinel system.

Response:

We appreciate the reviewer's recognition of our revision and also thank the instructive suggestion. The selective subtraction or separation of one/two transition metals from a mixture with multiple metals has always been one of the challenging problems. Similar chemical properties of those transition metals (Ni, Co, Mn) make it difficult to selectively isolate one with high efficiency. Here, we are inspired by the concept of Less is More, and find a subtractive reagent (ammonia oxalate) to preferentially isolate a certain proportion (0.1-mole ratio) of Co, which not only solves the precise subtraction of specific elements from the mixture but delivers an opportunity to directly reuse remaining to prepare next-generation 5 V class-Co-LNMO. The Co doping effects were revealed systematically, and the optimum proportion of Co doping was figured out. In addition, the problem of transition metal dissolution in LNMO was relieved, and high-temperature performance was improved. It is an effective modification way to synthesize LNMO with improved performance and is important to the sustainability of the LIBs industry.

In recent years, researchers have studied the separation of mixed metals but are limited by the use of many organic extractants which complicate the process, low

recovery rate of valuable elements, high cost, and poor controllability ¹⁻⁸

(Supplementary Table 1).

Table 1 Comparison of regular hydrometallurgical recycling methods from the reported works

Cathode material	Reagent	Product	Recovery rate
NCM ¹	H ₂ SO ₄ and H ₂ O ₂ , (C ₄ H ₈ N ₂ O ₂), P204 and C272	DMG Ni-(C ₄ H ₈ N ₂ O ₂) ₂ , MnO ₂ , CoC ₂ O ₄ and Li ₂ CO ₃	Ni (96.84%), Co (81.46%), and Mn (92.65%)
NCA ²	H ₂ SO ₄ , H ₂ O ₂ D2EHPA-Bis(2-ethylhexyl) phosphoric acid, Cyanex272®-Bis trimethylpentyl phosphinic acid	2,4,4- Co ₃ O ₄ (purity 83%), CoC ₂ O ₄ (purity 96%), NiO (purity 89%), and Li ₂ CO ₃ (purity 99.8%)	Co (80%~85%), Ni (90%) Li (72%).
NCM ³	Reduction roasting with starch, H ₂ SO ₄ , Mextral 984H (5-nonylsalicylaldoxime and 2-hydroxy-5-nonylaceto-phenone oxime),	No	Co (90.9%), Li (82%), Ni and Cu (98%)
NCM ⁴	KMnO ₄ , (trihexyltetradecylphosphoni-um	IL split-phosphinate molecule	extraction (as equilibria of Co ²⁺

m	bis(2,4,4-trimethylpentyl)phosphinate), aromatic solvent C10, NaOH	CoA ₂	and with ionic liquid stabilization of (>99%) phosphonium molecule by complexing with the chloride ions of aqueous solution (as R ₄ P ⁺ Cl ⁻) of the used ionic liquid (i.e., R ₄ P ⁺ A ⁻).
NCA ⁵	HCl, NaClO, NaOH	Co ₂ O ₃ ·3H ₂ O, Ni(OH) ₂	Co (90.25%) and Ni (96.36%), Li (>80%)
NCM ⁶	HCl, hypochlorite solution (NaClO), Na ₂ CO ₃	Mixture of manganese oxides (MnO ₂ , Mn ₃ O ₄ , Na _{0.55} Mn ₂ O ₄ ·1.5H ₂ O, CoCO ₃ , and Li ₂ CO ₃	Mn (95%) and Co (90%)
NCM ⁷	NH ₃ , (NH ₄) ₂ SO ₄ , Na ₂ SO ₃	(NH ₄) ₂ Mn(SO ₃) ₂ ·H ₂ O, Li ₂ SO ₄	NA, Leaching rate: Ni (94.8%),

		amine complexes	Co (88.4%), Mn (6.34%), and Li (96.7%)
NCM ⁸	(NH ₄) ₂ SO ₄ , (NH ₄) ₂ SO ₃	Li ₂ SO ₄ , (NH ₄) ₂ Co(SO ₄) ₂ · H ₂ O, (NH ₄) ₂ Mn(SO ₃) ₂ · H ₂ O (NH ₄) ₂ Mn(SO ₄) ₂ · 6H ₂ O	NA, Leaching rate: Ni (98%), Co (81%), Mn (92%), and Li and (98%)

The metal oxide with layer structure and stable chemical bonds in cathode materials makes it difficult to subtract proportionate elements without breaking down the whole cathode materials. The breaking of the chemical bond (e.g. TM-O bond) will lead to the distortion or even collapse of the crystal structure. It is impossible to achieve arbitrarily proportional chemical bond break without structural change. Therefore, the dissolution process to transform the metal element from the crystal structure to an ionic state in the solution may be requisite for subsequent separation and recycling at present. Though dissolution is commonly used in hydrometallurgy, the subsequent selective subtraction method in our work and the recycling route from waste to wealth have never been reported.

We select ammonium oxalate as a subtractive agent for the preferential subtraction of Co by regulation of pH and material feeding speed, which not only reduces the

requirement for scarce Co but also provides a solution and enlightenment for selective separation. The remaining can form a predetermined proportion of precursors with a very high reuse rate (>99.9%) of every transition metal element. However, the regular hydrometallurgy methods are for obtaining a single-component transition metal salt. A variety of precipitants/extractants even some hazardous and costly ones are required which involves a multi-step separation process for every single element (**Supplementary Table 1**), and the overall process is longer. The idea of Less is More in recycling can also inspire more research, thinking about how we implement selective subtraction considering scarcity and the high-value nature of metal elements.

From the perspective of economic analysis, suppose that 1 ton of ternary NCM523 spent batteries are treated and the regular hydrometallurgy strategy uses the common and inexpensive inorganic extractants, that the recovery rate is 100%, and that the recovered products are all transformed into commercial products without further purification. The processing steps involved in the subtractive strategy are fewer in contrast to the multi-step separation processes in the regular hydrometallurgy strategy. The material cost of the hydrometallurgy strategy is also higher than that of the subtractive strategy, as well as the profit, for which we have made systematically comparison in previous response letter. Based on the above analysis, we think that the proposed subtractive method and the concept are innovative and significant. We are going to study more green and straightforward subtraction methods in the following research. Thank you again for your suggestions and the ongoing discussion, which has played an important role in this paper and the subsequent research.

2) unless at higher temperatures, LNMO should cycle well, especially the pristine version. Why might the capacity fade so fast?

Response:

Thanks for your comment, the rapid capacity fading (i.e., “sudden death” in high-voltage operation or cycling at high temperature) may occur under aggressive but practically relevant conditions. Several major aging processes lead to capacity fading even a “Rollover”, as the following figure shows:

Fig. X2 Battery capacity fade. a) Schematic illustration of the gradual fade and the rollover failure. b) An overview of various battery degradation mechanisms. Wang, Y., Chang, X., Li, Z. *et al.* Preventing sudden death of high-energy lithium-ion batteries at elevated temperature through interfacial ion-flux rectification. *Advanced Functional Materials*. Copyright 2022 Wiley.

The electronic and ionic conductivities of pure LNMO are relatively poor compared with the Co-doped samples (reflected in **Fig. 1e**). The lower the conductivities of the materials, the more sluggish electron and ion diffusion ability. It shows poor compatibility between pure LNMO and the electrolyte during cycling,

which leads to forming worse interphase (shown in **Figs. 4-5**). In addition, the grain diameter of the primary particle of pure LNMO is smaller (see **Fig. S5c**), and there was more contact between the cathode particle and electrolyte, leading to more electrolyte decomposition with unfavorable interphase formation. In post-mortem analysis of LNMO after long-term cycling, the greater SEI resistances reflected from in-situ impedance changes of the cycled LMNO have verified that. When cycling at high temperatures (60°C), accelerated Li-ion flux leads to faster accumulation of forming unfavorable interphase due to the poor conductivities and interphase compatibility, which is different from the kinetically limited one at low temperatures. The exacerbation of interphase formation with high resistance accompanied by electrolyte decomposition make LNMO age faster and even “sudden death”.

Besides, the redox reaction, which can be carried out at high voltage (~ 4.7 V), allows the cycling of LNMO in the high voltage range for a long time. This scenario may exceed the limiting electrochemical window for ordinary electrolytes, leading to certain undesired electrolyte decomposition. The electrolyte we used is a common formula with EC : DMC : DEC (1:1:1 vol.) solvents, which lack high-voltage additives, and may result in more electrolyte decomposition even depletion under high temperatures. Compared to the reported work about LNMO, it is also shown that rapid capacity fades at high temperatures when cycling⁹. In contrast, the Co-doped LNMO has improved conductivities, and the better electron and ion transport properties provide better compatibility with electrolytes to form a more even and stable interphase under elevated temperature, it shows more stable cycling performance under aggressive but

practically relevant conditions.

On the other hand, there were more concerns about transition metal ion dissolution when cycling at high temperatures. The unstable interphase could not prevent the probability of dissolution, leading forming an unfavorable SEI composition with transition metal deposition. The surface of the Li anode in the cycled LNMO cell showed more byproducts (**Fig. 4d**), which further confirmed the formation of an unfavorable interface. Despite this deterioration, it will cause Li⁺ inventory depletion and/or impedance rise, so the capacity will suddenly decline sharply (*Journal of The Electrochemical Society*, 2017, 164 (2) A389-A399).

References

1. Li, C. *et al.* Separation and recovery of nickel cobalt manganese lithium from waste ternary lithium-ion batteries. *Separation and Purification Technology* **306**, 122559 (2023)
2. Atia, T. A., Elia, G., Hahn, R., Altimari, P. & Pagnanelli, F. Closed-loop hydrometallurgical treatment of end-of-life lithium ion batteries: Towards zero-waste process and metal recycling in advanced batteries. *Journal of Energy Chemistry* **35**, 220 (2019).
3. Yang, C., Zhang, J., Liang, G., Jin, H., Chen, Y. & Wang, C. An advanced strategy of “metallurgy before sorting” for recycling spent entire ternary lithium-ion batteries. *Journal of Cleaner Production* **361**, 132268 (2022).
4. Ilyas, S., Srivastava, R. R. & Kim, H. Selective separation of cobalt versus nickel by split-phosphinate complexation using a phosphonium-based ionic liquid. *Environmental Chemistry Letters* **21**, 673 (2023).
5. Joulié, M., Laucournet, R. & Billy, E. Hydrometallurgical process for the recovery of high value metals from spent lithium nickel cobalt aluminum oxide based lithium-ion batteries. *Journal of Power Sources* **247**, 551-555 (2014).
6. Barik, S., Prabakaran, G. & Kumar, L. Leaching and separation of Co and Mn from electrode materials of spent lithium-ion batteries using hydrochloric acid: Laboratory and pilot scale study. *Journal of Cleaner Production* **147**, 37-43 (2017).
7. Zheng, X. *et al.* Spent lithium-ion battery recycling—Reductive ammonia leaching of metals from cathode scrap by sodium sulphite. *Waste Management* **60**, 680-688 (2017).

8. Chen, Y., Liu, N., Hu, F., Ye, L., Xi, Y. & Yang, S. Thermal treatment and ammoniacal leaching for the recovery of valuable metals from spent lithium-ion batteries. *Waste Management* **75**, 469-476 (2018).
9. Dong, H. *et al.* Improved high temperature performance of a spinel $\text{LiNi}_{0.5}\text{Mn}_{1.5}\text{O}_4$ Cathode for high-voltage lithium-ion batteries by surface modification of a flexible conductive nanolayer. *ACS Omega* **4**, 185-194 (2019).